# Diacetyl Inhibits the Browning of Fresh-Cut Stem Lettuce by Regulating the Metabolism of Phenylpropane and Antioxidant Ability

**DOI:** 10.3390/foods12040740

**Published:** 2023-02-08

**Authors:** Xiaotong Li, Song Zhang, Qingguo Wang, Tiantian Dong

**Affiliations:** Key Laboratory of Food Processing Technology and Quality Control in Shandong Province, College of Food Science and Engineering, Shandong Agricultural University, Tai’an 271018, China

**Keywords:** stem lettuce slices, diacetyl fumigation, browning inhibitor

## Abstract

Enzymatic browning is the main quality issue of fresh-cut stem lettuce (*Lactuca sativa* L. *var. angustana Irish*). In this research, the effect of diacetyl on the browning and browning-related mechanisms of fresh-cut stem lettuce was explored. The data showed that diacetyl treatment with 10 μL L^−1^ inhibited the browning of fresh-cut stem lettuce and extended the shelf life by over 8 d at 4 °C compared with the control. Diacetyl treatment repressed gene expression and decreased the activities of PAL (phenylalanine ammonia-lyase), C4H (cinnamate-4-hydroxylase) and 4CL (4-coumarate-CoA ligase), which thus reduced the accumulation of individual and total phenolic compounds. Moreover, diacetyl enhanced the antioxidant ability and reduced ROS accumulation, improving the anti-browning ability and indirectly suppressing the biosynthesis of phenolic compounds. These results indicated that diacetyl treatment repressed the browning of fresh-cut stem lettuce by regulating the phenylpropanoid metabolism pathway and antioxidant ability. This study is the first to report that diacetyl has an effective anti-browning role for fresh-cut stem lettuce.

## 1. Introduction

Stem lettuce (*Lactuca sativa* L. *var. angustana Irish*), also called ‘Chinese lettuce’ or ‘asparagus lettuce’, is a popular vegetable that is consumed mainly in many countries for its rich nutrients, such as minerals, plant polysaccharides, bioactive components, etc. [1,2,3]. China is the largest country that produces and consumes stem lettuce products [2]. Stem lettuce has gained more attention for being processed as a ready-to-eat/fresh-cut product with the advantages of health, convenience and high nutrition reservation in recent years [3]. Nevertheless, fresh-cut stem lettuce is highly prone to browning on the cutting surface after being processed due to its abundant polyphenols and oxidases, which causes quality loss, and greatly reduces the commercial value and shelf life of the fresh-cut stem lettuce [4]. Currently, many physical and chemical methods, such as radio frequency, shortwave ultraviolet irradiation, ethanol, acetic acid combined with ethanol treatment, chlorine dioxide, allicin and 1-Methylcyclopropene (1-MCP), have been studied to inhibit the enzymatic browning of stem lettuce [5,6,7,8,9]. Though these methods had certain effects on inhibiting the surface discoloration of fresh-cut stem lettuces, few technologies were applied in the actual production for their cost, safety and inconvenient operation. Therefore, researchers are still working on exploring novel, safe, convenient and effectual strategies to suppress the enzymatic browning of fresh-cut stem lettuces to prolong the shelf life of products.

Diacetyl naturally occurs in many fruit aromas and fermented products as a volatile flavor compound, such as blackberry, lucuma, beer, wine, yogurt, bread, vinegar, etc. [10,11,12,13,14]. Synthetic diacetyl is widely used as a flavor enhancer in snack foods, bakery products, sauces and dairy products for its buttery flavor in the food industries [15]. Until now, there is no data suggesting that ingestion of diacetyl in foods is hazardous to human health, while the latest Code of Federal Regulations Title 21 (CFR) stated that diacetyl is still generally considered safe for its direct addition to human foods. Recent studies showed that diacetyl was capable of inhibiting the growth of fungal pathogens in postharvest grapes, mandarins, apples and strawberries and has the potential to be applied in the commercialization of postharvest fruit [16,17]. Moreover, diacetyl increased the phytohormone-mediated immunity ability and regulated the growth of plants [18,19]. Furthermore, diacetyl treatment eliminated the excessive reactive oxygen species (ROS), enhanced the disease resistance of plants, and thus alleviated the abiotic stress-induced senescence of *Arabidopsis* [18,20]. These results indicated that diacetyl could play a unique role in the immune regulation and physiological function of plants.

When the stem lettuce is cut and processed, tissue structures are destroyed, resulting in direct contact between phenolic substrates and oxidases, which triggers the enzymatic browning of fresh-cut stem lettuce. Moreover, the wounding stress will stimulate the generation and accumulation of ROS, causing oxidative damage to cells and tissues, which further accelerates cell wall decomposition and tissue browning [21,22]. However, the wounding stress and a large amount of ROS accumulation could activate the antioxidant enzymes, superoxide dismutase (SOD) and catalase (CAT), etc., to scavenge the excessive ROS [23,24]. In addition, the stress and excessive ROS could also induce the biosynthesis of phenolic compounds used for ROS neutralization, while this process could aggravate the browning via phenol oxidation [24,25]. In stem lettuce, the content level of phenolic compounds, especially chlorogenic acid, is significant for enzymatic browning [1,26]. The activity of phenylalanine ammonia-lyase (PAL) controls the synthesis of various phenolic compounds in the phenylpropanoid pathway of plants and plays a vital role in lettuce browning [27,28]. The identified PAL-related genes, *LsPAL1*, *LsPAL2*, *LsPAL3* and *LsPAL4*, are wound-inducible in lettuce [8,29]. Moreover, in addition to PAL, cinnamate-4-hydroxylase (C4H) and 4-coumarate-CoA ligase (4CL) also play a vital role in catalyzing the synthesis of phenolic compounds, and they are also induced by various biotic and abiotic stresses or ROS [24,30,31,32].

Given the above, we speculate that diacetyl treatment has the potential to inhibit the wound-induced browning of fresh-cut stem lettuces by regulating the phenylpropanoid pathway. However, few investigations are attainable on the influence of diacetyl on the enzymatic browning and browning-relevant physiological properties of fresh-cut fruit and vegetables. Therefore, the main objective of this research was to elucidate the biochemical and molecular mechanisms of the browning in fresh-cut stem lettuce influenced by diacetyl treatment and attempt to offer a theoretical foundation and viable technology for retarding the browning of fresh-cut stem lettuce.

## 2. Materials and Methods

### 2.1. Lettuce Material Preparation

The mature stem lettuces were obtained from a local market in Tai’an, Shandong, China, and then stored at 4 °C until they were used for further experiments. The stem lettuces with uniform sizes and shapes and free from diseases and mechanical damage were chosen for the experiments. Before cutting, the leaves were removed, and the stems were sanitized for 5 min with 200 μL L^−1^ sodium hypochlorite. Then, the stems were peeled with stainless steel knives. The peeled stem lettuces were washed with pre-cooled tap water, sanitized for 3 min with 50 μL L^−1^ sodium hypochlorite, drained and then cut into 5 mm thick slices using a stainless steel knife. Subsequently, the slices were sanitized, drained, dried with gauze and then used for diacetyl treatment.

### 2.2. Diacetyl Treatment

The 300 g of stem lettuce slices (100 g/bag, three bags/container) were selected randomly and placed evenly in a box (295 mm × 230 mm × 118 mm, Lock & Lock Co., Ltd., Seoul, Republic of Korea) that can be sealed completely. The stem lettuce slices were placed in air for the control, and the other stem slices were fumigated with different concentrations of diacetyl (1, 5, 10, 20 and 40 μL L^−1^) for 12 h at 4 °C, respectively. All diacetyl treatments in the experiment occurred in the ventilation equipment, according to our previous studies (1500 × 850 × 2350 mm). After fumigation, these stem lettuce slices were taken out and ventilated at 4 °C. Subsequently, the 100 g slices were packed into an LDPE (low-density polyethylene) plastic bag (175 × 250 × 0.04 mm). This experiment was set up with 7 evaluation time points. Each evaluation time point was prepared with 3 bags, and a total of 21 bags (three replicates/sample) were obtained per group (control and diacetyl-treated groups). Then, the bags were top-folded and stored at 4 °C. The color changes were recorded using images on days 0, 2, 4, 6, 8, 10 and 12 to confirm the optimal concentration of diacetyl to repress the browning of stem lettuce.

Likewise, the 300 g of stem lettuce slices were treated, which was consistent with the above treatment methods for comprehensively analyzing the mechanism of inhibiting the browning of fresh-cut stem lettuce by diacetyl; however, the concentration of diacetyl was 10 μL L^−1^ (the optimal concentration screened by the pre-experiment), and 0 μL L^−1^ was used as the control, similarly. On days 0, 2, 4, 6, 8, 10 and 12, during the storage of fresh-cut stem lettuce at 4 °C, 3 bags of fresh-cut stem slices (1 bag as one replicate) were taken out randomly for a visual and subjective browning evaluation, then immediately frozen in liquid nitrogen, ground into powder using a liquid nitrogen lapping machine, and stored at −80 °C for assaying the polyphenol synthesis, antioxidant-related enzymes and ROS accumulations.

In order to investigate the gene expression levels of the polyphenol biosynthesis-related enzymes, the stem lettuce slices were fumigated with and without (control) 10 μL L^−1^ diacetyl for 0, 1, 2, 4, 12 and 24 h. Then, the frozen stem lettuce slices were stored at −80 °C for RNA extraction and gene transcription analysis.

### 2.3. Overall Visual Quality Evaluation and Color Measurement of Fresh-Cut Stem Lettuce Slices

The overall visual quality of the fresh-cut stem lettuce slices was evaluated using a 9–1 point category test according to Chen et al. (2010) [33] and Peng et al. (2013) [34] with minor modification, where 9 = excellent, extremely fresh, none off-odor; 7 = good, marketable; 5 = fair, a limit of marketability; 3 = poor, a limit of usability; 1 = extremely poor, severe off-odor, unusable and spoiled. For each evaluation, twenty-four slices (eight from each replicate) were randomly selected and evaluated for the overall visual quality. Meanwhile, the color change in these slices was measured according to Peng et al. (2013) [34]. Readings were taken on five sites per side of each stem lettuce slice.

### 2.4. Assay of Total Phenols and Soluble Quinones

The content of total phenolics was detected by Meng et al. (2021) [35]. Briefly, frozen stem lettuce powder (2 g) was added to 10 mL of 70% acetone and homogenize. We then waited for 2 h at 25 °C for a sufficient reaction and centrifuged at 10,000× *g* for 20 min. Subsequently, the supernatant was used to assay the total phenolic.

For assaying the content of soluble quinones, three grams of frozen stem lettuce powder was added to 5 mL of methanol, and the measurement of soluble quinones referred to the description of Meng et al. (2022) [36].

### 2.5. PAL Activity Assay

The activity of PAL was assayed on the basis of Chen et al. (2017) [37]. Three grams of frozen stem lettuce powder (3 g) were added to 15 mL of 50 mmol L^−1^ borate buffer and then homogenized and centrifuged at 10,000× *g* for 15 min at 4 °C. Subsequently, the supernatant was used to determine the PAL enzyme activity at 290 nm. The PAL activity was expressed as U kg^−1^.

### 2.6. PPO Activity Assay

The PPO activity was analyzed by the procedure used according to Wang et al. (2015) [38]. Briefly, the frozen stem lettuce powder (2 g) was added to 8 mL of PBS with a pH of 6.8 and then centrifuged at 10,000× *g* for 15 min at 4 °C. Subsequently, the supernatant was used for the PPO activity assays. The activity of the PPO was determined at 410 nm and was expressed in U kg^−1^.

### 2.7. Activity Assays of 4CL and C4H

The activities of C4H and 4CL were assayed following a method of Jiang et al. (2019) [39]. In brief, the frozen stem lettuce powder (2 g) was added to 8 mL of 0.1 mol L^−1^ Tris-HCl buffer with a pH of 8.7 and then centrifuged at 12,000× *g* for 30 min at 4 °C. Subsequently, the supernatant was used for testing the activities of C4H and 4CL.

For the C4H activity assay, the supernatant (100 μL) was homogenized with 1.4 mL of 0.1 mol L^−1^ Tris-HCl reaction solution. Subsequently, the solution was placed at 25 °C for 30 min to react sufficiently. Then, the reaction was stopped by adding 50 μL of 6 mol L^−1^ HCl and 75 μL of 6 mol L^−1^ NaOH. The absorbance was recorded at 340 nm. The C4H activity was expressed in U kg^−1^.

For measuring the 4CL activity, 50 μL of the supernatant was added into 950 μL of 0.1 mol L^−1^ Tris-HCl solution. Then, the solution was placed at 40 °C for 10 min to react adequately, and it was terminated by adding 0.1 mL of 50 μmol L^−1^ HCl to this mixture. The changes in absorbance were recorded at 333 nm. The 4CL activity was expressed as U kg^−1^.

### 2.8. Determination of Phenolic Substance Content

The determinations of chlorogenic acid, caffeic acid, p-coumaric acid and ferulic acid were used by HPLC. They were extracted on the basis of the procedure by Wang et al. (2015) [38] and Feng et al. (2020) [40]. Five grams of stem lettuce powder was homogenized with 20 mL of formic acid with HPLC grade and incubated for 24 h at 25 °C. The mixture was filtered and washed repeatedly with formic acid. The filtrate was concentrated by vacuum rotary evaporation at 55 °C. Then, the concentrated solution was mixed with 15 mL of ethyl acetate and continued to evaporate at 37 °C until nearly all the fluids were evaporated. Subsequently, the methanol with HPLC grade (2 mL) was employed for dissolving the residue and then filtered through 0.45 μm. Afterward, the solution was used to detect the phenolic substances. The subsequent HPLC analysis was on the basis of the description by Feng et al. (2020) [40].

### 2.9. In Vitro Experiments Exploring the Effect of Diacetyl on PAL Activity

The inhibition of diacetyl on the PAL activity in vitro was conducted based on the description by Huang et al. (2020) [8]. The diacetyl was added to the PAL enzyme preparation at a final concentration of 10 μL L^−1^ and stored for 0, 1, 4, 12 and 24 h to measure the PAL activity on the basis of Section 2.5.

### 2.10. Total Antioxidant Capacity

#### 2.10.1. ABTS Inhibition Rate

Briefly, frozen stem lettuce powder (2 g) was added to 5 mL of 95% ethanol. Subsequently, the ABTS scavenging ability of stem lettuce was based on Meng et al. (2021) [35].

#### 2.10.2. H_2_O_2_ Content and O_2_^•−^ Production Rate

The frozen stem lettuce powder (1 g) was placed into the 15 mL centrifuged tube and mixed with 3 mL of pre-cooled acetone. Then, the mixture was swirled and shocked immediately. The H_2_O_2_ determination referred to the description by Hu et al. (2012) [41]. For assaying the production rate of O_2_^•−^, briefly, stem lettuce powder (2 g) was added to 6 mL of phosphate buffered solution, and after homogenization, the mixture was centrifuged at 4 °C, 8000× *g* for 10 min. Then, the supernatant was employed for the analysis of the production rate of O_2_^•−^, based on Gao et al. (2016).

#### 2.10.3. CAT Enzyme Activity

The activity of CAT was measured, referring to the description by Peng et al. (2013) [34]. Briefly, the frozen stem lettuce powder (2 g) was added into 10 mL of 0.1 mol L^−1^ PBS and centrifuged at 4 °C, 12,000× *g* for 15 min after homogenization. Subsequently, the supernatant was employed for testing CAT activity and was expressed in U kg ^−1^.

### 2.11. RNA Extraction and RT-qPCR Analysis

The extraction of total RNA from the stem lettuce slices was conducted using the Fastpure Universal Plant Total Isolation Kit (Vazyme, Nanjing, China), according to the paper’s introduction. The reverse transcription of RNA was determined by the description of Li et al. (2022) [42]. The concentrations were measured by Nano Drop2000 (Thermo Fisher Scientific, Waltham, MA, USA).

The primer sequences used for the quantitative analysis in this study referred to Huang et al. (2020) [8] and Liu et al. (2022) [43]. The synthesis of all primer sequences and the analysis of the RT-qPCR are on the basis of the method by Li et al. (2022) [42].

### 2.12. Statistical Analyses

All data from this experiment were statistically analyzed and expressed as the mean ± standard deviation (SD). The charts used in this study were produced in the GraphPad Prism 8.3.0 software, and the significance of all data was determined using the SPSS 23.0 software for analysis (*p* < 0.05) [42].

## 3. Results

### 3.1. Effect of Different Concentrations of Diacetyl on the Browning of Fresh-Cut Stem Lettuce Slices

The visual appearance of the fresh-cut stem lettuce slices treated with different concentrations of diacetyl is shown in Appendix A. The surface color of the control and 1 μL L^−1^ diacetyl-treated stem lettuce slices seriously browned, and the color of the stem lettuce slices treated with 5 μL L^−1^ diacetyl turned slightly brown on day 12 at 4 °C. However, the fresh-cut stem lettuces treated with 10, 20 and 40 μL L^−1^ diacetyl exhibited a superior appearance with green coloration and little discoloration (Appendix A). Considering the safety in application, the minimal concentration of 10 μL L^−1^ diacetyl was therefore employed to verify its effect on the browning inhibition of fresh-cut stem lettuce and further investigate the related mechanisms in the following experiments. Figure 1A shows the visual quality of the control stem lettuce slices and the slices treated with 10 μL L^−1^ diacetyl for 12 h. The control slices were severely browned on day 12 during storage, whereas the stem lettuce slices treated with diacetyl did not brown at all, which illustrated that diacetyl alleviated the browning of stem lettuce slices. Figure 1B shows that the diacetyl treatment maintained the overall visual quality of the stem lettuce slices compared to the control. The control slices had lost commercial value (visual quality score ≤ 5) with a visual quality score below five on day 4 during storage after cutting, while the diacetyl-treated stem lettuce slices were still marketable because the senor score was above seven on day 12 (Figure 1B). Furthermore, diacetyl treatment maintained higher values of L* and b* (Figure 1C,D) and suppressed the rise of a* value during all the storage progress (Figure 1E), reflecting that diacetyl treatment alleviated the browning of lettuce slices, which coincided with the differences in visual color between the control and diacetyl treatment. All the above results demonstrated that diacetyl treatment delayed the enzymatic browning and prolonged the shelf life of fresh-cut stem lettuce for more than 8 d at 4 °C.

### 3.2. Diacetyl Treatment Inhibited Phenolics Generation and Quinone Synthesis

Phenolic compounds can be oxidized and converted into quinines, which are key factors that influence the enzymatic browning of stem lettuce [1,26]. Hence, the changes in total phenolics, total quinones and four main individual phenolic compounds in stem lettuce slices were determined first. As shown in Figure 2A,B, the contents of the total phenolics and total quinones in the control fresh-cut stem lettuces both increased after cutting during storage time, while they had no significant changes until day 6 during storage in diacetyl-treated stem lettuce slices (Figure 2A,B). Moreover, the contents of both the total phenolics and total quinones in the control stem lettuces were far higher than those in diacetyl-treated stem lettuce at the later stage of storage (Figure 2A,B). It suggested that diacetyl treatment might inhibit the production of total phenolics, decrease the synthesis of brown pigment quinones, and thus reduce the browning of fresh-cut stem lettuces. Meanwhile, the individual phenolic compounds in stem lettuce slices were measured, as shown in Figure 2C–F. After cutting, chlorogenic acid and caffeic acid in control stem lettuce synthesized rapidly at the early stage of storage (from day 0 to day 4), and their contents varied slowly after day 4 during storage. On the other hand, the generations of chlorogenic acid and caffeic acid in diacetyl-treated stem lettuce were lower compared with that in control stem lettuce (Figure 2C,D). Furthermore, the generations of *p*-coumaric acid and ferulic acid in control stem lettuce were increasing continually, especially at the late stage of storage after cutting. Additionally, the two individual phenolics were less increased in diacetyl-treated stem lettuce, and their contents were much lower than that in control stem lettuce in the whole progress of storage (Figure 2E,F). These results indicated that diacetyl could suppress the synthesis of individual phenolic compounds, which decreased the content of total phenolics and the generation of total quinones (Figure 2A,B).

### 3.3. Diacetyl Treatment Reduced Activities and Gene Expression Levels of PAL

The activity of PAL controls the synthesis of phenolic compounds, and its activity can be induced by wound stress [24]. PAL activity in control stem lettuce increased after cutting, almost by 4.25 times from day 0 to day 6. However, the activity in diacetyl-treated stem lettuce slices decreased after diacetyl treatment and rose much more slowly after day 2 in comparison with that in control stem lettuce slices (Figure 3A). It showed that the diacetyl treatment was able to well-repress the increase in PAL activity in stem lettuce slices. To verify the inhibiting effect of diacetyl on PAL activity, the PAL crude enzyme extracted from stem lettuce was treated with diacetyl. Interestingly, diacetyl did not inhibit the activity of the PAL crude enzyme in vitro (Figure 3B), which indicated that diacetyl could not directly affect PAL activity by interacting with the protein of PAL. Therefore, the transcript levels of *PAL* genes, *LsPAL1*, *LsPAL2*, *LsPAL3* and *LsPAL4* in fresh-cut stem lettuce treated with diacetyl for 1 h, 4 h, 12 h and 24 h were further measured, respectively. The expression levels of all the PAL genes showed increasing trends after cutting in control stem lettuce, while the expressions of *LsPAL1*, *LsPAL2*, *LsPAL3* and *LsPAL4* were all suppressed after diacetyl treatment for 1 h, and also treatment for 4 h, 12 h and 24 h (Figure 3C–F). These above results indicate that diacetyl could repress the transcript levels of *PAL* genes at an early stage and thus reduce PAL activity in fresh-cut stem lettuce.

### 3.4. Diacetyl Treatment Repressed Activities and Gene Expression of C4H and 4CL

C4H and 4CL also play a vital role in the synthesis of phenolic compounds. C4H catalyzes the formation of cinnamic acid into p-coumaric acid and subsequently generates caffeic acid, which further produces ferulic acid. Furthermore, *p*-coumaric acid can also be transformed into chlorogenic acid under the catalysis of 4CL and other enzymes [31]. Therefore, the transcript level and activities of the two enzymes were determined here. The results showed that the activities of 4CL and C4H had no significant differences between the control and diacetyl-treated stem lettuce slices on day 0 after diacetyl treatment (Figure 4A,B). With storage time increasing, the activities of 4CL and C4H were increased in both the control and diacetyl-treated stem lettuce. However, the activities of 4CL and C4H in control stem lettuce increased much more rapidly than that in diacetyl-treated stem lettuce slices (Figure 4A,B). Moreover, the expression levels of 4CL and C4H in stem lettuce slices were all reduced after diacetyl treatment (Figure 4C,D). The above results suggested that diacetyl treatment repressed the gene expression of 4CL and C4H and thus reduced their enzyme activities in the stem lettuce slices.

### 3.5. Diacetyl Improves Antioxidant Capacity of Fresh-Cut Stem Lettuce Slices

Wounding stress can induce excessive ROS of fresh-cut fruit and vegetables, which accelerates cell disruption and aggravates browning. Meanwhile, wounding stress can also activate antioxidant systems to eliminate ROS and retard oxidative damage [21,44]. ABTS^+^ is a stable free radical of fruit and vegetables, and its ability to scavenge free radicals is a common method to measure the total antioxidant capacity [45]. Figure 5A shows that the ABTS inhibition rate increased first and decreased in both the control and diacetyl treatment. Nevertheless, the ABTS inhibition rate in diacetyl-treated stem lettuce slices was improved compared with the control slices, suggesting that diacetyl treatment increased the antioxidant ability of fresh-cut stem lettuce. Furthermore, as storage time increased, CAT activity in diacetyl-treated stem lettuce increased after treatment and had the highest value on day 6 during storage. Nevertheless, CAT activity in the control stem lettuce slices remained nearly constant during all the storage time (Figure 5B). CAT activity in diacetyl-treated stem lettuce was higher than that in control stem lettuce slices after diacetyl treatment, on the whole (Figure 5B). Furthermore, the reactive oxygen H_2_O_2_ and O_2_^•−^ were determined, and the results showed that H_2_O_2_ and O_2_^•−^ were both increased in control stem lettuce, especially H_2_O_2_, where the content had a quick rush from day 2 to day 4 (Figure 5C,D). The contents of H_2_O_2_ and O_2_^•−^ in diacetyl-treated stem lettuce were lower than that in control stem lettuce, though they had an increasing process during storage (Figure 5C,D). The results suggested that diacetyl treatment improved the antioxidant abilities, increased the capacity of eliminating ROS, and thus reduced the accumulation of ROS in stem lettuce, which contributed to alleviating the browning of stem lettuce.

## 4. Discussion

As a natural food-flavoring additive, diacetyl has been considered as GRAS by FEMA (www.femaflavor.org, accessed on 10 December 2022), and the ingestion of diacetyl in food directly is safe for humans. Nowadays, diacetyl has been widely used as a food flavoring for buttery fragrances. Furthermore, the content of diacetyl in yogurt is 200–3000 mg kg^−1^ [46]. The dose of diacetyl used in this study was 10 μL L^−1^, about 164 mg kg^−1^ fresh-cut stem lettuce slices for fumigation, and the residual value in stem lettuce was much lower than the used dose. Previous studies reported that diacetyl might cause respiratory illness in workers exposed to it in microwave popcorn manufacturing facilities [47]. However, in the following practical application, automation can be achieved during diacetyl treatment in a closed environment, which could prevent workers from contacting diacetyl directly and decrease the harmful effect on workers’ health in the best possible way. Therefore, the residual dose in stem lettuce and the processing procedure should be safe for humans.

Browning is recognized as a necessary element, affecting the quality of fresh-cut lettuce [5,33], which is mainly caused by the oxidation of phenolic compounds and influenced by the membrane stability associated with the metabolism of the phenylpropanoid pathway and the antioxidant ability to scavenge ROS [48]. Diacetyl treatment alleviated the browning of stem lettuce in this study. The control stem lettuce turned red-brown on day 12 during storage time at 4 °C, while the diacetyl-treated stem lettuce still had better visual quality (Figure 1A). As we all know, the oxidation of polyphenols is catalyzed by polyphenol oxidase (PPO) and is the reason for the browning of some fruit and vegetables [49]. However, the PPO activities in fresh-cut stem lettuce between groups with and without diacetyl treatment had no significant difference during storage time (Appendix A). Meanwhile, some reports have shown that there is no direct relationship between PPO activity and lettuce browning, and the browning of lettuce is closely relevant to PAL activity [27,28,50,51]. Heat shock or mild heat treatment (50–60 °C) suppressed the cut lettuce enzymatic browning of cut lettuce by inhibiting wound-induced PAL [50,52]. Huang et al. (2020) [8] explained that PAL is the major element for the browning of the butt in lettuce instead of PPO. These results were consistent with our results that the browning of fresh-cut stem lettuce is closely related to PAL activity rather than PPO activity.

PAL controls the transformation of phenylalanine into cinnamic acid as the rate-limiting enzyme in the phenylpropanoid pathway, which is subsequently catalyzed by C4H to form *p*-coumaric acid. On the one hand, *p*-coumaric acid can be transformed into chlorogenic acid under the catalysis of 4CL and other enzymes. On the other hand, *p*-coumaric acid can be turned into caffeic acid and ferulic acid following a series of reactions [31]. Four genes encoding PAL in lettuce, including *LsPAL1*, *LsPAL2*, *LsPAL3* and *LsPAL4*, can be induced by wounding, and acetic acid treatment could suppress the gene expression of *LsPALs* and retard the browning of lettuce [29]. Similarly, we found that PAL activity was increased (Figure 3A), and the expression of *LsPALs* was enhanced in control stem lettuce after cutting (Figure 3C–F), which is in line with previous studies [8,29]. However, the increase in PAL activity was completely repressed throughout the storage time, and the expression of PAL-related genes was inhibited by diacetyl treatment (Figure 3C–F). Furthermore, the activities and gene expression of 4CL and C4H were also reduced after diacetyl treatment (Figure 4). The decrease in activity and transcript levels of all these enzymes would result in a lower generation of total and individual phenolics in the diacetyl-treated stem lettuce, which is demonstrated by our results (Figure 2). All the above results demonstrated that the diacetyl treatment repressed the gene expression and activities of phenolic synthesis-related enzymes, decreased the production of phenolic compounds and thus reduced the browning of fresh-cut stem lettuce.

Furthermore, wounding stress induces the generation and accumulation of ROS, which could disrupt cell membranes and accelerate the enzymatic browning of lettuce [50,53,54]. Wounding can activate antioxidant enzymes such as CAT to scavenge free radicals [55]. This research demonstrated that the diacetyl treatment improved the activity of CAT and maintained a higher free radical-eliminating capacity, which reduced the generation of H_2_O_2_ and O_2_^•−^ (Figure 5). Moreover, ROS was correlated to the activation of phenylpropanoid metabolism [32]. Wounding and excessive ROS can induce and activate phenolic-related enzymes and increase the generation of phenolic compounds, which helps to resist wounding stress and neutralize excessive ROS [23,24,56,57]. It was consistent with our results that phenolic-related enzymes, PAL, 4CL and C4H, and phenolic contents were increased in control stem lettuce after cutting. However, diacetyl has the potential to scavenge ROS [18], which might reduce the induction of gene expression and activities of phenolic-related enzymes and thus decrease the biosynthesis of phenolics, which was in line with our results (Figure 2, Figure 3 and Figure 4). Therefore, based on our results and previous studies, we have summarized the mechanisms by which diacetyl treatment represses the browning of stem lettuce slices [21,29,30,32,42,58]. As shown in Figure 6, wounding stress activates the enzyme activities that are relevant to phenylpropanoid metabolism and promotes the biosynthesis of phenolic compounds. Moreover, wounding stress accelerates ROS accumulation, which also induces and activates the phenolic synthesis-related enzymes and thus facilitates phenylpropanoid metabolism. However, diacetyl treatment could repress gene expression and decrease the activities of the related enzymes associated with phenylpropanoid metabolism, thus reducing phenolic compounds. Furthermore, diacetyl treatment improved the ROS scavenging enzyme activities and inhibited the generation of ROS, which indirectly suppressed the biosynthesis pathway of phenolic compounds and ultimately inhibited the browning of fresh-cut stem lettuce. Nevertheless, how diacetyl affects ROS accumulation and phenylpropanoid metabolism needs further research.

## 5. Conclusions

In this present study, diacetyl treatment with 10 μL L^−1^ inhibited the browning of fresh-cut stem lettuce and extended the shelf life during storage at 4 °C. Diacetyl treatment repressed gene expression and decreased the activities of enzymes relevant to the phenylpropanoid metabolism pathway, such as PAL, C4H and 4CL, which thus reduced the accumulation of individual and total phenolic compounds. Furthermore, diacetyl improved the antioxidant capacity and reduced the accumulation of H_2_O_2_ and O_2_^−^, which indirectly suppressed the biosynthesis of phenolic compounds and enhanced the ability of anti-browning in stem lettuce. The results suggested that diacetyl treatment alleviated the browning of stem lettuce, mainly by inhibiting the metabolism of the phenylpropanoid pathway and decreasing the synthesis of phenolic compounds.

## Figures and Tables

**Figure 1 foods-12-00740-f001:**
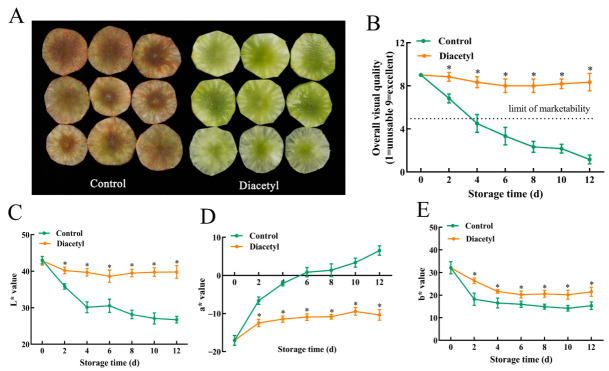
Effect of 10 μL L^−1^ diacetyl treatment on the browning of stem lettuce slices. (**A**) Visual browning photographed on day 12 during storage at 4 °C. (**B**–**E**) Variations in overall visual quality (**B**), L* (**C**), a* (**D**) and b* (**E**). Asterisks * indicate the significant differences (*p* < 0.05) between control and treated groups at the same storage time. Vertical bars represent standard deviations.

**Figure 2 foods-12-00740-f002:**
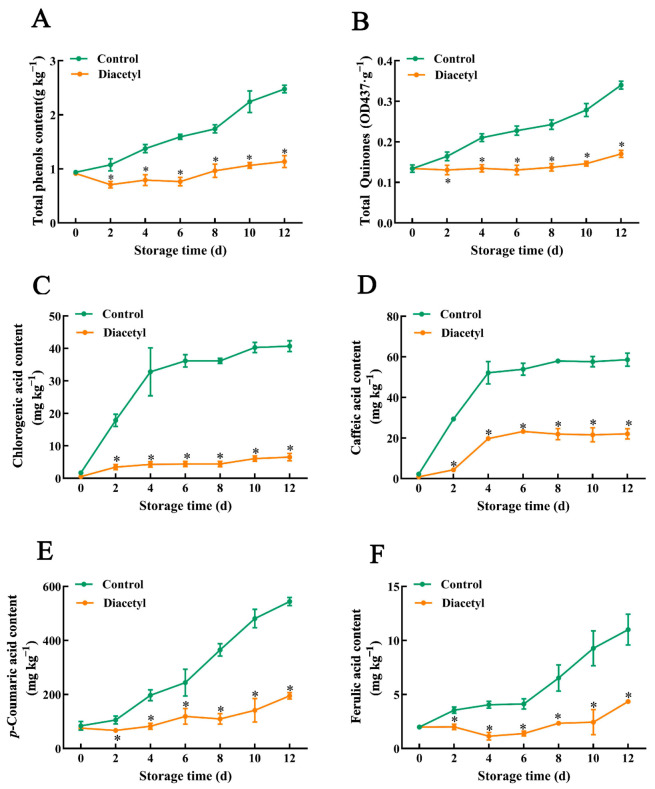
Changes of total phenols (**A**), total quinones (**B**) and individual phenols (**C**–**F**) in fresh-cut stem lettuce slices during storage time at 4 °C after diacetyl treatment. Asterisks * indicate the significant differences (*p* < 0.05) between control and treated groups in the figure at the same storage time. Vertical bars represent standard deviations.

**Figure 3 foods-12-00740-f003:**
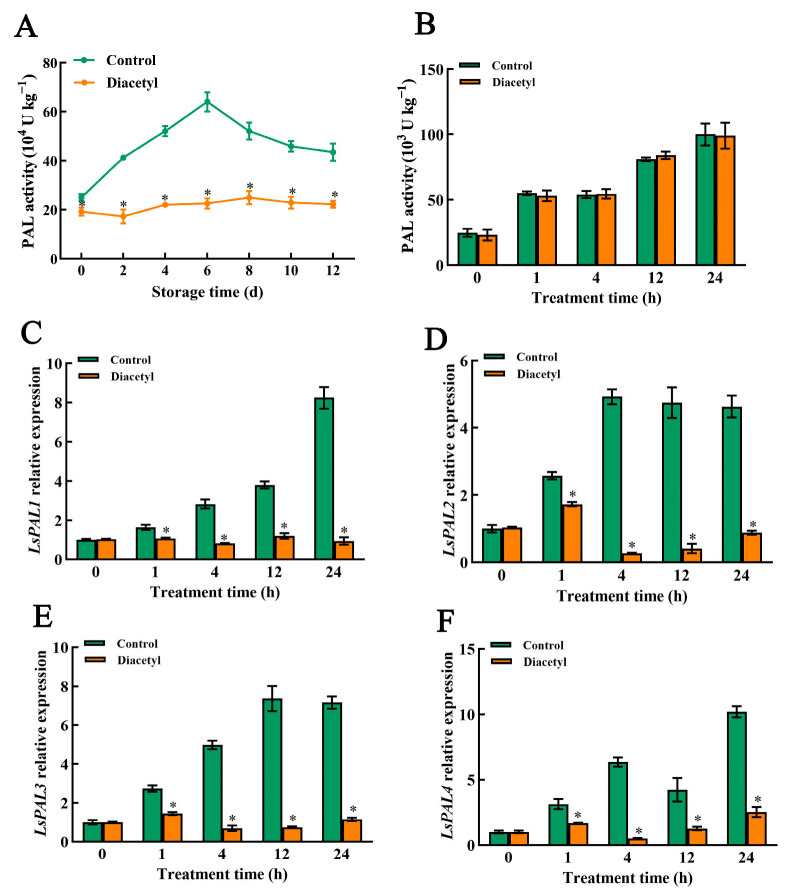
Diacetyl caused a great reduction in activities and expression levels of phenolic-related enzymes and genes. PAL activity in vivo (**A**), PAL activity in vitro (**B**), the gene expressions of *LsPAL1* (**C**), *LsPAL2* (**D**), *LsPAL3* (**E**), *LsPAL4* (**F**) of fresh-cut stem lettuce at 4 °C. Asterisks * indicate the significant differences (*p* < 0.05) between control and treated groups in the figure at the same storage time. Vertical bars represent standard deviations.

**Figure 4 foods-12-00740-f004:**
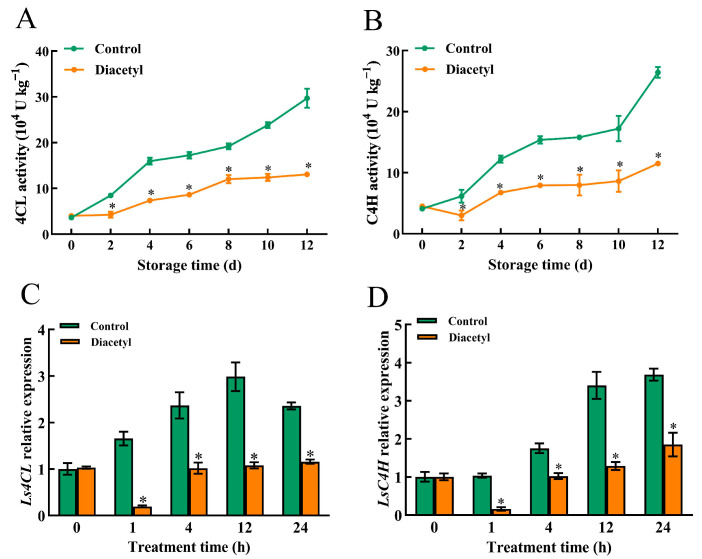
Diacetyl reduced the activities and gene expressions of 4CL and C4H. 4CL activity (**A**), C4H activity (**B**), the gene expressions of *Ls4CL* (**C**) and *LsC4H* (**D**) of fresh-cut stem lettuce at 4 °C. Asterisks * indicate the significant differences (*p* < 0.05) between control and treated groups in the figure at the same storage time. Vertical bars represent standard deviations.

**Figure 5 foods-12-00740-f005:**
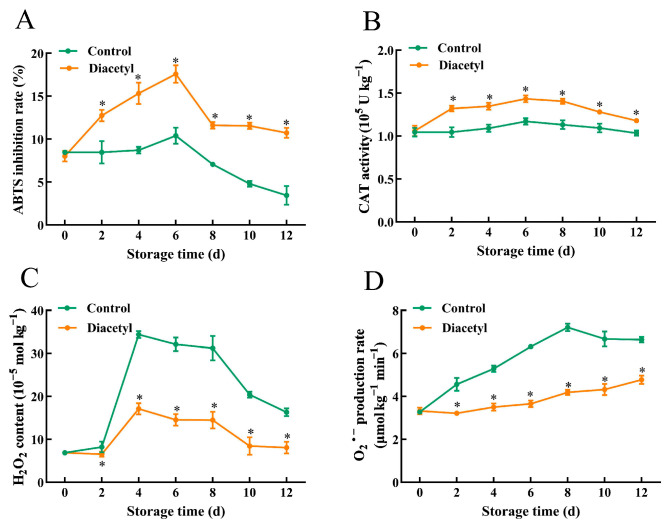
Diacetyl improved the antioxidant capacity of fresh-cut stem lettuce slices during storage time at 4 °C. (**A**) ABTS inhibition rate. (**B**) CAT activity. (**C**) H_2_O_2_ content and O_2_^•−^ content (**D**). Asterisks * indicate the significant differences (*p* < 0.05) between control and treated groups in the figure at the same storage time. Vertical bars represent standard deviations.

**Figure 6 foods-12-00740-f006:**
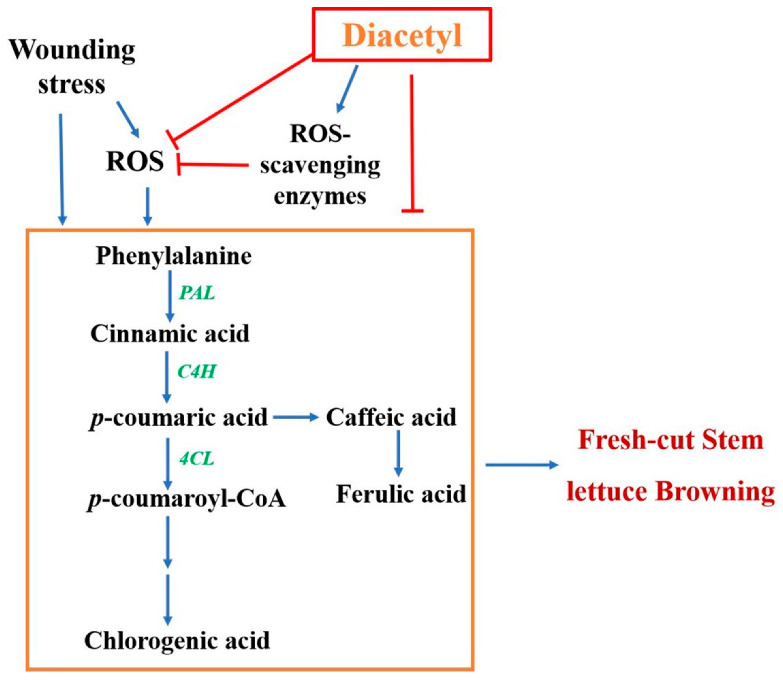
The diagrammatic model of the potential mechanism of diacetyl treatment regulating the browning of fresh-cut stem lettuce. The red lines represent a negative effect, while the blue lines represent a positive effect.

## Data Availability

The data supporting the results of this study are all included in the present article.

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
