# Peer review of "Diacetyl Inhibits the Browning of Fresh-Cut Stem Lettuce by Regulating the Metabolism of Phenylpropane and Antioxidant Ability"

_foods, 2023, doi:10.3390/foods12040740_

Round 1
Reviewer 1 Report
The main purpose of this paper, according to authors, was to show that diacetyl treatment is capable of repressing the browning of fresh-cut stem lettuce via regulating the phenylpropanoid metabolism pathway and antioxidant ability. In my opinion, this article deals with an interesting topic and a promising pre-treatment, that could alleviate a serious quality problem of fresh cut fruits and vegetables. The introduction is well written and the experimental procedure adequately explained and detailed. Furthermore, diagrams succeed in demonstrating clearly the results obtained and the discussion part provides clear justifications on the main findings. An issue that is of major importance is the safety of diacetyl application, as authors also describe in lines 367-378. It is not clear which dosage is allowed for food applications, and whether workers’ safety is already addressed in food industry. Is that pretreatment currently applied? Besides lettuce, is it also applied for other tissues?
Regarding other points that need clarifications/corrections are the following:
Line 12:…was explored..
Line 32: omit the article ‘the’ in the phrase ‘Nevertheless, THE fresh-cut stem..’
Line 33/59: besides cutting, does any other processing step takes place?
Line 40: which techniques were actually applied? What were the disadvantages and why a new method (diacetyl fumigation) is necessary? When those ‘conventional’ techniques are applied, what is the shelf life extension obtained (compared to the untreated, ‘control’ samples)?
Line 44: …compound, which is..
Line 50/51 (see also general comments): is there any legislation limit of diacetyl concentration/dosage for food applications?
Line 57: replace the word ‘took’ with ‘can take’
Line 60:…triggers..
Line 77: ..potential to inhibit..
Line 88: for this type of lettuce, which is the lowest temperature suggested to avoid chilling injury? Is 4C an advisable storage temperature? Please, comment on this.
Lines 87-95: why are 3 sanitization steps applied? Is that a common industrial practice (or a way to exclude any microbial contamination)?
Line 100: in my opinion, there should be another group of samples, where a conventional anti-browning treatment should be applied, so as to be able to correctly compare the ‘proposed’ diacetyl technique.
Line 101: why for 12 h fumigation? Explain the choice of the experimental conditions.
Lines 100-108/lines 110-116: what is the difference between these two experimental procedures? Not well described. How did you choose between the different diacetyl concentration to move to the next experiment? Try to be more explicit and clear regarding the experimental design.
Line 152: ..by a method used in Wang et al.,..
Line 190: ..was used to detect..
Line 236: change title of the sub-section to ‘Effect of diacetyl concentration on the..’
Line 239:..are shown..
Line 253: how did you decide that the value of ‘5’ is the sensory limit?
Line 260: this extension obtained is based merely on color deterioration (browning)? Could any other spoilage (from a microbiological aspect) be also the limiting factor? Which other degradation mechanisms could be involved in lettuce deterioration?
Line 268: do you mean ‘quinones’ (instead of quinines)?
Line 282: Instead of ‘whereas’ write ‘on the other hand’
Line 285: Instead of ‘while’ write ‘Additionally’
Line 313:…during subsequent storage or also immediately after diacetyl treatment (time zero for storage)?
Line 320: the word ‘Moreover; can be omitted.
Line 348: replace the word ‘went by’ with ‘increased’
Figure 5A (y-axis title: ABTS inhibition rate)
Line 386: ‘As is well known’
Line 399: On one hand,…
Check the formatting (different sizes of letters) in lines 427-443 and 455-456.
Author Response
Please see the attachment, thanks.

Reviewer 2 Report
Review Foods 2154652
Diacetyl inhibits the browning of fresh-cut stem lettuce via regulating the phenylpropane metabolism and antioxidant ability
The paper discusses the use of diacetyl in the prevention of tissue browning. The work extensively discusses the obtained in the experiment and proves the effectiveness of this application.
The second part of the title should be:
….regulating the metabolism of methylpropane and the antioxidant ability.
Key words: please discard the words used in the title.
Line 27: … vegetable mainly consumed in many countries for… → vegetable that is consumed mainly in many countries..
Line 47: for prolonging → to prolong
Lines 49-50: Till now, there is no data suggesting that ingestion of diacetyl in foods is hazardous to human health and the latest….
Your statement is exaggerated. Please conduct better research on the harmfulness of use Diacetyl. First of all why Diacetyl is banned for use in e-cigarettes in EU and USA?.
Another example: P. Kovacic, A.L. Cooksy, Electron transfer as a potential cause of diacetyl toxicity in popcorn lung disease, „Reviews of Environmental Contamination and Toxicology”, 204, 2010, s. 133–148, DOI: 10.1007/978-1-4419-1440-8_2.
This two examples I found searching only one minute.
Line 53: potential to apply → potential to be applied
Line 55: Moreover, diacetyl treatment scavenged → Furthermore, diacetyl treatment eliminated
Lines 77-83: Since one of the goals of the work is to investigate the molecular and biochemical role of diacetyl, there should be more information about the chemical structure of diacetyl in the introduction. The chemical structure allows to predict what types of reactions may occur in the plant after the use of this chemical compound.
Lines 87-95: →The mature →market till→untill
Lines 94, 143, 200: Afterward → Subsequently
Line 103: → removed
Line 109: optimum → optimal
Line 120: To research → to investigate
Lines 130-132: For every evaluation, twenty-four slices … were selected randomly → For each evaluation 24 slices… were randomly selected
Line 158: was regarded – was considered
Line 166: for assaying activities → for testing activities
Line 202: was used for assaying ABTS scavenging ability → was used to test the ABTS scavenging ability
Line 217: was regarded → was considered
Line 223: and the cDNA synthesis used → and the synthesis of cDNA was performed using
Lines 240-266: Why only the results of the CIE Lab are presented. Many color experiments prove that the color angle h0 = (tan-1 b*/a*) and the ratio a"'/b*, because literature data show that they can be better color indicators than individual coefficients.
Line 297: by wounding stress → by wound stress
Line 302: the rise on → the increse in
Line 380: human beings → humans
Line 384: ability of scavenging ROS → ability to eliminate ROS
Line 394: suppressed enzymatic browning of cut lettuce via inhibiting the induction of wound-induced PAL → suppressed cut lettuce enzymatic browning of cut lettuce via inhibiting wound-induced PAL
Author Response
Please see the attachment, thanks.

Reviewer 3 Report
The manuscript describes the effect of diacetyl on the control of browning of fresh-cut stem lettuce. The treatment with diacetyl affected PAL synthetic pathway, ROS accumulation and the expression of PAL enzymes as well as the activity of antioxidant enzymes.
The objectives are clearly defined, and the results are consistent with them. The manuscript is well organized and presented, however some minor points need further attention to improve the manuscript.
-Line 48 I would remove “in food industries”
-Line 50 please add a reference
-Line 51 food is uncountable
-Paragraph 2.2 please describe the ventilation equipment
-Paragraph 2.4 please indicate how do you express the results of total phenolics and soluble quinones
-Line 143 powder
-Line 144 please indicate the procedure employed to homogenize the mixture of borate buffer and sample.
-Lines 146-147 please change into: All the reaction mixtures were incubated at 40 C° for 1h, and added with 0.1 mL of 6 mol L-1 to stop the reaction.
-Line 153 pH
-Line 155 were
-Line 166 was containing?
-Line 170 please indicate at what pH the reaction mixture was adjusted
-Line 210 please indicate how the results were expressed
-Line 216 why did you use the plural? How many homogenizations have been carried out?
-Line 228 “The Ultra SYBR Mixture Kit 227 (Cowin Biosciences, Beijing, China) was used for RT-qPCR analysis”
-Line 239 in figure S1 there is no indication of the days of storage, please add it.
-In figure 1B maybe the graph would be more complete if you add a line indicating the limit of marketability
-Line 268 quinones?
-Line 272 delete were
-Line 320 I would avoid starting a new paragraph with moreover
- from line 427 onwards please uniform character size.
Author Response
Please see the attachment, thanks.
